# Exploiting Representation Curvature for Boundary Detection in Time Series

**Yooju Shin**[1], **Jaehyun Park**[1], **Hwanjun Song**[1], **Susik Yoon**[2], **Byung Suk Lee**[3], **Jae-Gil Lee**[1]*

[1]KAIST, [2]Korea University, [3]University of Vermont

{yooju.shin, jhpark813, jaegil, songhwanjun}@kaist.ac.kr, susik@korea.ac.kr, bslee@uvm.edu

## Abstract

*Boundaries* are the timestamps at which a class in a time series changes. Recently, representation-based boundary detection has gained popularity, but its emphasis on consecutive distance difference backfires, especially when the changes are gradual. In this paper, we propose a boundary detection method, **RECURVE**, based on a novel change metric, the ***curvature*** of a representation trajectory, to accommodate both gradual and abrupt changes. Here, a sequence of representations in the representation space is interpreted as a trajectory, and a curvature at each timestamp can be computed. Using the theory of random walk, we formally show that the mean curvature is lower near boundaries than at other points. Extensive experiments using diverse real-world time-series datasets confirm the superiority of RECURVE over state-of-the-art methods.

## 1 Introduction

In a time series composed of sequential data points (simply points) indexed by timestamps, there are *boundaries* (or *change points*) signifying transitions between different classes or states, such as a shift from running to walking [1, 2]. Detecting boundaries is a crucial task in preprocessing and diverse applications of time-series data. As preprocessing, they partition a time series into segments of coherent points, accelerating annotation of the time-series for further analysis and giving additional supervision in classification [3, 4, 5, 6, 7]. As primary tasks, they are valuable for identifying changes that require human attention in a variety of domains, including climate, health care, finance, and manufacturing; epilepsy detection, stock price tracking, and action segmentation are examples of possible applications [8, 9, 10, 11].

Representation-based boundary detection methods [12, 13] are prevalent today because they do not require specific assumptions on time-series properties, such as distribution or temporal shape, and can handle high dimensionality due to the capability of a self-supervised model that autonomously learns distinctive features from raw time series without any supervision. In these methods, a self-supervised model [14, 15, 16] is first used to derive a representation of each point, and then a point is identified as a boundary if its representation significantly deviates from those of adjacent points. Let's refer to the points close to a boundary as *inter-segment* points and the remaining points as *intra-segment* points. In short, these methods operate by assuming that the *distance* between consecutive representations is greater between inter-segment points than between intra-segment points.

However, this assumption on the distance difference does *not* always hold, especially when the changes are subtle or gradual. Time-series representation learning methods often pursue preserving the *temporal coherence* of a time series as their training goal is to make temporally close points similar in their representations and distant points dissimilar [14]. As demonstrated in Figure 1, the consecutive distances are not clearly distinguishable between intra- and inter-segment points for relatively subtle changes with `stair up` ↔ `stair down` because just the direction of motion differs

---

*Corresponding author.

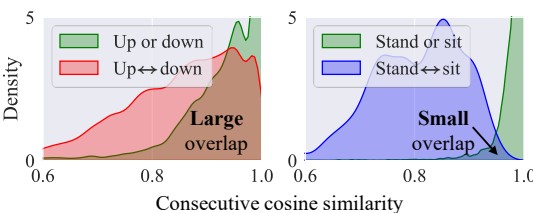
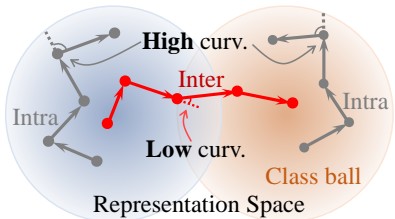

Figure 1: Consecutive distance (cosine similarity) distribution from intra- and inter-segment points in the representations of the HAPT dataset.

Figure 2: Curvature comparison between intra- and inter-segment points in a representation space.

between the two classes, whereas they are for abrupt changes with `stand` ↔ `sit`. Thus, the inability to handle subtle changes hinders achieving an overall good performance.

In this work, we take a novel perspective on detecting boundaries by leveraging ***curvatures*** instead of distances in the representation space. As shown in Figure 2, the curvature at a point in a curve measures the instantaneous rate of direction change, or more precisely, the amount by which the curve deviates from being a straight line [17]. Suppose that a sequence of point representations from a time series constitutes a *representation trajectory*. We observe that, regardless of whether the changes are gradual or abrupt, the direction of the representation trajectory tends to change *more sharply* (showing a *higher curvature*) at intra-segment points than at inter-segment points. Accordingly, we contend that the curvature of a representation trajectory should be a very promising indicator for class boundary detection.

Using Figure 2, we justify the intuition behind *curvature-based* boundary detection. Because representation learning tries to learn class-separated features, well-embedded points of a certain class (or a segment) can be drawn from its class-specific ball [18, 19]. That is, the representation trajectory of intra-segment points is confined within a class ball, whereas that of inter-segment points is not. Then, for intra-segment points to reside exclusively within a class ball, their representation trajectory needs to make sharp turns frequently. In contrast, the transition from one class ball to another does not necessarily make sharp turns. This observation is formally proven by the relationship between the mean curvature and the radius of a confining hypersphere, assuming a random walk of a point representation (see Section 3.4).

Overall, for boundary detection, treating a sequence of representations as a trajectory and measuring its curvature is an entirely novel approach, which results in RECURVE (Representation trajEctory CURVaturE). A representation trajectory is derived by a time-series representation learning method, and the curvature at each point is calculated very efficiently; then, the points whose curvature is relatively small are identified as boundaries. RECURVE is simple yet powerful, and can be combined with any time-series representation learning method. We conduct comprehensive evaluations on a variety of time-series datasets, comparing it against state-of-the-art boundary detection methods. The results demonstrate that RECURVE consistently enhances the accuracy, achieving improvements of up to 12.7%. Furthermore, this superiority is shown to exist regardless of the degree of change between different classes.

## 2 Related Work

### 2.1 Boundary Detection

Time-series boundary detection methods assess the dissimilarity between two successive intervals and apply a threshold to pinpoint the positions of boundaries. There are multiple methods available for quantifying dissimilarity: **(1)** conducting statistical tests, **(2)** quantifying the deviation from discovered patterns, and **(3)** calculating distances between the representations learned from a self-supervised model. We summarize each category here, with additional in-depth details available in extensive surveys [1, 2].

Statistical tests often rely on the probability density ratio of two consecutive intervals as a key statistic. CUSUM is a traditional parametric algorithm that adds up the log likelihood ratio when a probability density function is given [20, 21]. RuLSIF is a non-parametric algorithm that directly estimates the probability density ratio using Pearson divergence without a probability density function [22, 23, 24]. A kernel-based statistical test maps each interval to a kernel space and then computes the kernel

Fisher discriminant ratio as a statistic [25]. KL-CPD uses a deep neural network as a generator for kernel parameters, which solves high sensitivity in selecting parameters [26].

The proactive discovery of frequent temporal patterns is necessary for temporal pattern-based boundary detection. FLOSS stores the locations of similar subsequences in a time series using Matrix Profile and measures the likelihood of a regime change [27]. Motif-based boundary detection relies on the identification of short temporal patterns (motifs) determined through the minimum description length criterion; these motifs are then compared for similarity with other subsequences within a time series [11, 28]. ESPRESSO, on the other hand, is a hybrid of pattern- and statistic-based approaches, detecting a wide range of boundaries across different scenarios and data types [29].

Representation-based boundary detection methods are distinguished by the manner in which a self-supervised model is trained. TIRE exploits an autoencoder to retain time-invariant features in consecutive timestamps to make representations of boundaries salient [12]; after training, the output representations undergo a process of smoothing, wherein a moving average is applied prior to the dissimilarity computation. TS-CP$^2$ leverages contrastive learning techniques to promote close proximity between representations of two consecutive timestamps and distant proximity between representations at randomly selected timestamps [13]; it examines the difference between each consecutive distance and the moving average.

## 2.2 Time-Series Representation Learning

Time-series representation learning builds a model to create versatile representations capable of performing diverse downstream tasks such as classification, forecasting, and anomaly detection [30, 31]. Reconstruction-based learning methods train autoencoder-based deep neural networks using a reconstruction loss. TimeNet is an early example that uses a sequence-to-sequence autoencoder and uses the hidden embedding extracted from the encoder as a representation [32]. DTCR extends traditional reconstruction-based learning by incorporating a $k$-means loss alongside the reconstruction loss [33]. Input masking is also commonly used for reconstructing data with specific timestamps intentionally masked or hidden [34, 35, 36].

In contrastive learning, the Info-NCE (Noise Contrastive Estimation) loss plays a pivotal role by bringing a positive pair closer together and pushing a negative pair apart in the representation space. An early approach considers a sampled window and a subsequence from the window as a positive pair [37]. In recent methods such as TNC (Temporal Neighborhood Coding), the temporal distance serves as a criterion for identifying a positive pair, keeping two neighboring timestamp representations close [13, 14, 38]. Following the principles of SimCLR [39], a positive pair can be created by pairing a sampled window with its augmentation which involves data perturbation or context changes [15, 40]. Besides, the Fourier transform of a time series serves as an augmentation technique for generating positive pairs or providing a new representation space [16, 41, 42].

## 3 RECURVE: Curvature-based Boundary Detection

### 3.1 Preliminaries and Problem Setting

**Dataset and Model:** Let $\mathcal{X} = (\mathbf{x}_t)_{t=1}^T$ be a time series, where $T$ is the total number of points, and $\mathbf{x}_t \in \mathbb{R}^d$ is a $d$-dimensional point at timestamp $t$. Let $\mathcal{C} = \{t_k \mid k \in [\![1, K]\!]\}$ be a set of the timestamps for the ground-truth boundaries. Considering class labels annotated at each timestamp, $\mathcal{C}$ is composed of the timestamps where there is a change in the label from the previous one (e.g., stand $\rightarrow$ walk). A window $X_{t_m} = (\mathbf{x}_t)_{t=t_m-I}^{t_m+I-1}$ is a sequence of consecutive $2I$ points centered at timestamp $t_m$. A representation model $f_\theta$, which is a deep neural network parameterized by $\theta$, converts each window $X_{t_m}$ to its *representation* $\mathbf{z}_{t_m} \in \mathbb{R}^{d'}$, i.e., $\mathbf{z}_{t_m} = f_\theta(X_{t_m})$.

**Representation Learning:** RECURVE is not bound to a specific representation learning method, and we summarize the training process using one of the popular methods, the *temporal predictive coding (TPC)* proposed in TS-CP$^2$ [13]. Here, two non-overlapping consecutive windows are used as a positive pair, and two randomly-sampled windows are used as a negative pair. Thus, TS-CP$^2$ randomly samples $b$ windows as well as their succeeding windows and constructs a batch $B = \{X_{t_1}, X_{t_2}, \ldots, X_{t_b}, X_{t_1+2I}, X_{t_2+2I}, \ldots, X_{t_b+2I}\}$; the method subsequently minimizes the InfoNCE loss $\ell_{\text{NCE}}$ [43],

$$\ell_{\text{NCE}} = -\frac{1}{b}\sum_{j=1}^{b}\log\frac{\exp(\text{sim}(\mathbf{z}_{t_j}, \mathbf{z}_{t_j+2I})/\tau)}{\sum_{k=1,k\neq j}^{b}\exp(\text{sim}(\mathbf{z}_{t_j}, \mathbf{z}_{t_k})/\tau)}, \tag{1}$$

where $\text{sim}(\cdot, \cdot)$ is the cosine similarity function, $\exp(\cdot)$ is the exponential function, $\mathbf{z}_{t_j} = f_\theta(X_{t_j})$, and $\tau$ is a scaling parameter. The model parameter $\theta$ is updated iteratively by gradient descent, i.e., $\theta \leftarrow \theta - \eta\nabla_\theta\ell(B, \theta)$, where $\eta$ is a learning rate.

**Change Metric and Detection:** Using the representations of all windows centered at each point in $\mathcal{X}$, i.e., $\{\mathbf{z}_t \mid t \in [\![1, T]\!]\}$, a *change metric* $\hat{y}_t$ is derived for each point $\mathbf{x}_t \in \mathcal{X}$, which represents the extent that $\mathbf{x}_t$ is a boundary. For example, the change metric in TS-CP$^2$ employs the *distance* (i.e., cosine similarity) between the embeddings of adjacent points,

$$\hat{y}_t^{\text{dist}} = \text{NORM}(|\text{sim}(\mathbf{z}_t, \mathbf{z}_{t+1}) - \text{MA}(\text{sim}(\mathbf{z}_t, \mathbf{z}_{t+1}))|), \tag{2}$$

where MA calculates a simple central moving average and NORM is min-max normalization over all timestamps to rescale a value between 0 and 1. Then, similar to binary classification, the points whose change metric exceeds a certain threshold $\varphi$ are identified as boundaries,

$$\hat{\mathcal{C}} = \{t \mid \hat{y}_t \geq \varphi \text{ where } t \in [\![1, T]\!]\}. \tag{3}$$

**Goal:** Obviously, an effective change metric is crucial to the success of boundary detection. Therefore, we propose a novel change metric, $\hat{y}_t^{curv}$, using the *curvatures in the representation space* instead of the consecutive distances in the representation space.

## 3.2 Curvature-Based Change Metric

A trajectory usually refers to the path or track that an object (e.g., human and vehicle) in motion follows through space and time [44]. Thus, we get to Definition 3.1 if we think of an object as a point floating in the representation space.

**Definition 3.1** (TRAJECTORY). *A **representation trajectory** (simply **trajectory**) $\mathcal{T}$ is a curve specified by a sequence of representations at consecutive timestamps and denoted as $\mathcal{T} = (\mathbf{z}_t)_{t=1}^{|\mathcal{T}|}$.*

The curvature at a specific point on a curve is the rate at which the direction of the curve changes instantaneously at the point [17]. It is a well-defined concept in geometry and quantifies how sharply or gradually the curve bends or deviates from a straight line. We employ the definition designed for a trajectory [45]. For three timestamps in order, $t^-$, $t$, and $t^+$, where $t^- < t < t^+$, consider their representations $\mathbf{z}_{t^-}$, $\mathbf{z}_t$, and $\mathbf{z}_{t^+}$.

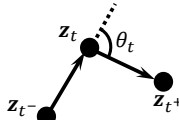

Figure 3: Turning angle.

Two difference vectors, $\mathbf{z}_t - \mathbf{z}_{t^-}$ and $\mathbf{z}_{t^+} - \mathbf{z}_t$, are naturally derived, and the *turning angle* $\theta_t$ between them in Figure 3 is calculated by

$$\theta_t = \arccos\frac{(\mathbf{z}_t - \mathbf{z}_{t^-}) \cdot (\mathbf{z}_{t^+} - \mathbf{z}_t)}{||\mathbf{z}_t - \mathbf{z}_{t^-}||\,||\mathbf{z}_{t^+} - \mathbf{z}_t||}. \tag{4}$$

Each value of $\theta_t$ ranges between 0 and $\pi$, where $t \in [\![2, |\mathcal{T}|-1]\!]$. Then, the *curvature* is the rate of the direction changes between the two difference vectors, i.e. how much a difference vector rotates per unit length, as defined in Definition 3.2.

**Definition 3.2** (CURVATURE). *The **curvature** at timestamp $t$ in a representation trajectory $\mathcal{T}$ is the turning angle $\theta_t$ divided by the sum of the difference vector lengths,*

$$\kappa_t = \frac{\theta_t}{||\mathbf{z}_t - \mathbf{z}_{t^-}|| + ||\mathbf{z}_{t^+} - \mathbf{z}_t||}. \tag{5}$$

According to our observations and intuitions described in Section 1, the curvature of an intra-segment point is higher than that of an inter-segment point. Thus, the curvature defined in Definition 3.2 can be used as a change metric. The computational complexity remains $O(d'T)$, consistent with that of TS-CP$^2$ where cosine similarity is computed instead of curvature. For stability, the timestamps $t^-$ and $t^+$ in Eq. (5) are determined to be $w \geq 1$ timestamps before and after timestamp $t$, i.e., $t^- = t - w$ and $t^+ = t + w$. We set $w$ to 5% of the mean segment length, which is observed to work well in most situations. Please refer to Section 4.6 about the sensitivity analysis on the value of $w$. Definition 3.3 concludes our novel curvature-based change metric.

**Definition 3.3** (CHANGE METRIC). *The **curvature-based change metric** at timestamp $t$ becomes*

$$\hat{y}_t^{curv} = \text{MA}(1 - \text{NORM}(\kappa_t)), \tag{6}$$

where $\kappa_t$ is obtained from Eq. (5) and MA and NORM are the same as Eq. (2).

In Definition 3.3, we normalize the curvature to a scale of 0 to 1. Subtracting this normalized curvature from 1 ensures that the change metric increases as the curvature decreases while maintaining the scale. Finally, a moving average is employed to smooth the curvature and mitigate fluctuations.

## 3.3 Change Metric Thresholding

Once the change metric $\hat{y}_t^{curv}$ in Eq. (6) is prepared, it is possible to detect boundaries by finding the points where $\hat{y}_t^{curv} \geq \varphi$, as formulated by Eq. (3). Therefore, it is necessary to develop a heuristic for determining the threshold $\varphi$, and additional information can be utilized for this purpose. Such additional information includes the mean segment length and the validation dataset. If the mean segment length, i.e., the average of the lengths of segments distinguished by boundaries, is known, the estimated number of boundaries can be calculated by dividing the total number of timestamps by the mean segment length. The threshold $\varphi$ is then determined to obtain the estimated number of boundaries. Alternatively, if we have a validation dataset, we select the threshold $\varphi$ that yields the best performance based on an evaluation measure. Empirical evaluation in Section 4 employs the mean segment length in thresholding.

## 3.4 Theoretical Analysis

Our theoretical analysis is conducted by showing the following properties: **(1)** the intra-segment points in the representation space are confined within a smaller hypersphere than the inter-segment points, as shown in Figure 4; **(2)** the mean curvature of a representation trajectory increases as the radius of the confining hypersphere decreases, which leads to the rationale behind Definition 3.3.

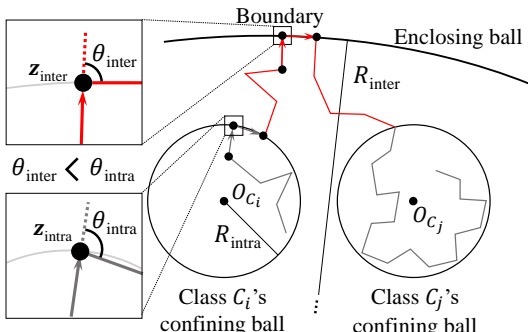

Figure 4: Comparison of the curvatures between intra- and inter-segment points.

**Definition 3.4** (CONFINEMENT). Consider a subsequence of a specific class $C_i$, $\mathcal{X}_{C_i} = (\mathbf{x}_t)_{t=t_{start}}^{t_{end}}$, as well as its representation trajectory, $\mathcal{T}_{C_i} = (\mathbf{z}_t)_{t=t_{start}}^{t_{end}}$, in Figure 4. Then, $\mathcal{T}_{C_i}$ is **confined** within a hypersphere $S_{C_i} \subset \mathbb{R}^{d'}$ centered at $O_{C_i} \in \mathbb{R}^{d'}$ of radius $R_{intra}$ if and only if $||\mathbf{z}_t - O_{C_i}|| < R_{intra}$ holds for all $t \in [\![t_{start}, \ t_{end}]\!]$.

Definition 3.4 comes from a widely known fact that representation (contrastive) learning produces *class-separated* representations [18, 19]. The augmented positive examples form a connected graph based on augmentation overlap; thus, the alignment of positive examples by contrastive learning will cluster the examples of the same class together and lead to class-separated representations [18].

**Proposition 3.5** (CONFINEMENT RADIUS). *Consider a transition from a class $C_i$ to another class $C_j$ in Figure 4. Let $S_{C_i}, S_{C_j} \subset \mathbb{R}^{d'}$ be the confining hyperspheres for $C_i$ and $C_j$, respectively, of radius $R_{intra}$. Then, consider a larger hypersphere of radius $R_{inter}$ that encloses the inter-segment points (in red) as well as $S_{C_i}$ and $S_{C_j}$. Thus, $R_{intra} < R_{inter}$ holds by definition.*

Based on temporal coherence [7, 13, 46] inherent in time series, we make an assumption on the representation trajectory before proceeding to the second step.

**Assumption 3.6** (EQUILATERAL RANDOM WALK). A representation trajectory $\mathcal{T} = (\mathbf{z}_t)_{t=1}^{|\mathcal{T}|}$ is a Markov chain, where $\mathbf{z}_t$ is sampled over the surface of the *unit* hypersphere centered at $\mathbf{z}_{t-1}$ and also contained in a confining hypersphere of radius $R$. That is, $||\mathbf{z}_t - \mathbf{z}_{t-1}|| = 1$ ($t \in [\![2, \ |\mathcal{T}|]\!]$) and $||\mathbf{z}_t|| < R$ ($t \in [\![1, \ |\mathcal{T}|]\!]$) such that $R > 1$.

Under Assumption 3.6, the curvature in Eq. (5) becomes the turning angle in Eq. (4) because the denominator is reduced to a constant when $w = 1$. Then, when a given representation trajectory $\mathcal{T}$ is confined by a hypersphere of radius $R$, its *mean curvature* is defined by

$$\mathcal{K}_{\mathcal{T}}(R) = \frac{1}{|\mathcal{T}|} \sum_{\mathbf{z}_t \in \mathcal{T}} \mathbb{E}_{\mathbf{z}_t | R}[\theta_t], \tag{7}$$

where $\mathbb{E}_{\mathbf{z}_t | R}[\theta_t]$ is the expectation of the curvature at timestamp $t$ with respect to the distribution of the representations in the confining hypersphere of radius $R$.

**Lemma 3.7** (MEAN CURVATURE). *Consider a representation trajectory $\mathcal{T}$ confined in a hypersphere of radius $R$ under Assumption 3.6. Then, the mean curvature $\mathcal{K}_\mathcal{T}(R)$ is a **decreasing** function of the radius $R$, i.e., $\frac{d}{dR}\mathcal{K}_\mathcal{T}(R) < 0$.*

The proof of Lemma 3.7 is provided by Diao et al. [47]. The mean curvature is rigorously formulated as a complicated integral. By a simulation of random walk with one million steps, the decrease in the curvature is represented by the linear function $3.53 - 1.21R$ and the function $\pi/2 + 0.65/R^{1.5}$ for two different regimes of $R$.

**Notation.** The representation trajectories confined within the hyperspheres of radii $R_{intra}$ and $R_{inter}$ in Figure 4 are called *intra-segment* and *inter-segment* trajectories as well as denoted by $\mathcal{T}_{intra}$ and $\mathcal{T}_{inter}$, respectively.

Putting Proposition 3.5 and Lemma 3.7 together, the observation on the difference in the curvature is finally formalized by Theorem 3.8.

**Theorem 3.8** (CURVATURE DIFFERENCE). *The mean curvature of an intra-segment trajectory is greater than that of an inter-segment trajectory, i.e., $\mathcal{K}_{\mathcal{T}_{intra}}(R_{intra}) > \mathcal{K}_{\mathcal{T}_{inter}}(R_{inter})$.*

*Proof.* Because $R_{intra} < R_{inter}$ by Proposition 3.5, $\mathcal{K}_{\mathcal{T}_{intra}}(R_{intra}) > \mathcal{K}_{\mathcal{T}_{inter}}(R_{inter})$ obviously holds by the decreasing nature of $\mathcal{K}_\mathcal{T}(R)$ of Lemma 3.7. $\square$

Theorem 3.8 can be intuitively explained if we consider the special case in which the next representation of $\mathbf{z}_{intra}$ or $\mathbf{z}_{inter}$ lies on the surface of a hypersphere, as visualized in Figure 4. Since a smaller radius necessitates a sharper turn, $\theta_{intra} > \theta_{inter}$ holds true. In this particular instance, where $\mathbf{z}_{intra}$ or $\mathbf{z}_{inter}$ is an orthogonal projection onto the surface, the turning angle can be expressed as $\pi - \arccos\frac{1}{2R}$, a *decreasing* function of $R$. Please refer to Appendix A for more details.

### 3.5 Empirical Validation

The findings in the theoretical analysis also align well with the visualizations of the representations from a real dataset. Figure 5 displays three representation trajectories in the representation space of *two* principal components, which are obtained by the TPC method with $d' = 32$ for the mHealth dataset. Each representation trajectory includes 100 points centered at a boundary, where each point is sampled every ten points in the original trajectory. Inter-segment points within five sampled timestamps from the boundary are denoted by "×", while intra-segment points are denoted by "●". The color of each symbol indicates the value of our change metric—i.e., $1-$curvature. Obviously, inter-segment points have higher values of the change metric than intra-segment points. Interestingly, in Figure 5, the distance between two consecutive representations remains similar regardless of whether they are intra- or inter-segment points. This result reaffirms the existence of temporal coherence in the representation space, which could reduce the accuracy of *distance-based* methods. Moreover, it is evident that the representation trajectories of intra-segment points exhibit clearer confinement, resulting in more closed shapes and larger average turning angles. The representation trajectories of inter-segment points have fewer rotations and produce a relatively straighter shape.

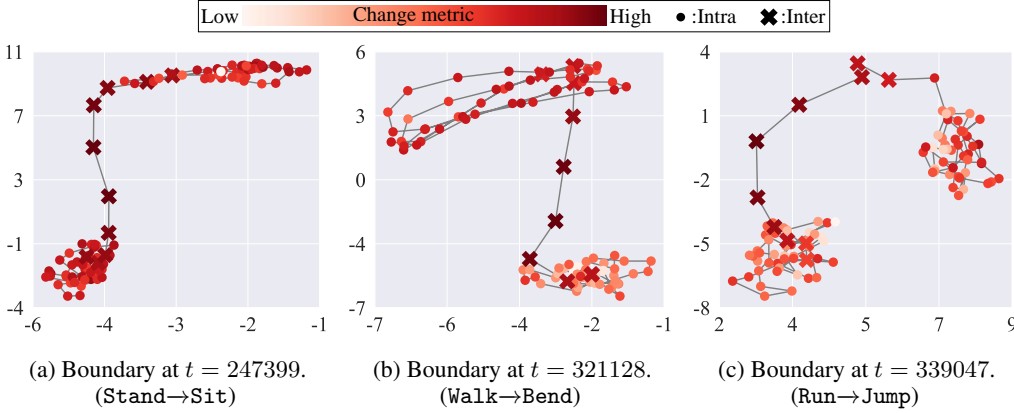

(a) Boundary at $t = 247399$.
(Stand→Sit)

(b) Boundary at $t = 321128$.
(Walk→Bend)

(c) Boundary at $t = 339047$.
(Run→Jump)

Figure 5: Three representation trajectories in the space of two principal components in mHealth.

# 4 Evaluation

## 4.1 Experiment Setting

**Datasets:** The profiles of the four datasets used in our experiments are summarized in Table 1, which lists the number of timestamps, mean segment length, number of classes, data dimensionality, sampling rate in Hz, and number of boundaries. WISDM[48],

Table 1: Summary of datasets and hyper-parameters.

| Dataset | Timestamps | Length | Class | $d$ | Rate | #Bo | Window | Epoch |
|---------|-----------|--------|-------|-----|------|-----|--------|-------|
| WISDM | 343092 | 697 | 6 | 3 | 20 | 491 | 50 | 10 |
| HAPT | 407807 | 903 | 6 | 6 | 50 | 450 | 100 | 50 |
| mHealth | 343195 | 2932 | 12 | 23 | 50 | 119 | 100 | 50 |
| 50salads | 496250 | 551 | 19 | 2048 | 30 | 898 | 50 | 100 |

HAPT[49], and mHealth[49] are human action recognition datasets, which are measured by single or multiple accelerometers and/or gyroscopes. 50salads[50] is a video dataset that captures 25 people preparing salads; the I3D features of 2048 dimensions are extracted, following Farha and Gall[3]. The set of ground-truth boundaries, $\mathcal{C}$, is defined as the set of the timestamps where the class changes. The dimensionality of the representation space is set to $d' = 8$ for WISDM and HAPT and $d' = 32$ for mHealth and 50salads, considering their data dimensionality.

**RECURVE Details:** To obtain the point representations, we employ two time-series representation learning methods, TPC proposed in TS-CP$^2$[13] and TNC[14]. RECURVE+TPC and RECURVE+TNC indicate the two implementations depending on the representation learning method. A temporal convolutional network (TCN) is trained in both methods. Note that any representation learning method can be combined with RECURVE. The window size, $2I$, and the number of training epochs for each dataset are shown in Table 1, where the window size is approximately twice the sampling rate. The learning rate is set to $0.005$ for all datasets. The hyperparameter $w$, indicating the length of a representation vector, is set to $5\%$ of the mean segment length. After obtaining a change metric for each timestamp, in Eq. (6), the same normalization is applied to all windows from an individual dataset. The moving average in Eq. (6) is computed using the ten timestamps preceding and following each timestamp. RECURVE is implemented using PyTorch 1.13.0, and its source code is available at `https://github.com/kaist-dmlab/RECURVE`.

**Compared Methods:** RuLSIF[24], KL-CPD[26], and TS-CP$^2$[13] are chosen as the representative method from each of the three categories in Section 2.1. The window size in Table 1 is applied to all compared methods for fair comparison. A multilayer perceptron is used for the regressor of RulSIF. The hyperparameters of RuLSIF and KL-CPD are favorably determined by a grid search, as detailed in Appendix B. The public implementations of RuLSIF$^2$ and KL-CPD$^3$ are used for our experiments. TS-CP$^2$ is the closest to our work, and its main mechanism is briefly described in Section 3.1. Because representation learning itself is shared between TS-CP$^2$ and RECURVE when TPC is used, the same hyperparameter setting is applied to both methods whenever possible. TS-CP$^2$ is re-implemented using PyTorch 1.13.0 for direct comparison with RECURVE.

**Evaluation Measures:** First, the Area Under the ROC Curve (AUC) is measured by considering boundary detection as binary classification with a binary label vector $\mathbf{y} \in \{0, 1\}^T$ converted from $\mathcal{C}$. Following Deldari et al.[13], an error margin is introduced to accommodate some noise from annotation and detection. A detected boundary is considered to be correct if it lies within $p$ timestamps from one of the ground-truth boundaries. For this purpose, $\mathbf{y}$ is relaxed to

$$y_t = \begin{cases} 1 & \text{if } t_k - p \leq t < t_k + p \text{ where } t_k \in \mathcal{C} \\ 0 & \text{otherwise.} \end{cases} \tag{8}$$

Then, for $t \in [\![1, T]\!]$, whether ($\hat{y}_t$ in Eqs. (2) or (6) $\geq \varphi$) is compared against $y_t$ in Eq. (8). We use multiple error margins, $p \in \{5, 10, 20\}$, since a margin could be different for diverse applications[1]. Second, the mean LOCation distance (LOC) is measured, which is the average distance from a detected boundary to its closest ground-truth boundary[27, 51]. The LOC measure is useful for checking the preciseness of the boundaries detected.

Regarding the threshold $\varphi$, the AUC measure does not require a specific value because it evaluates the true positive and false positive rates over a given range. For the LOC measure, two values are used for each experiment: one is determined to achieve the best F1 score, and the other is determined

---

$^2$`https://github.com/HSE-LAMBDA/roerich`
$^3$`https://github.com/HolyBayes/klcpd`

by the heuristic based on the mean segment length in Section 3.3, where the estimated number of boundaries is multiplied by $p = 10$, taking the error margin into account.

For each evaluation measure, we conduct every experiment *five* times with different seeds and report the average as well as the standard deviation. We use Intel(R) Xeon(R) Gold 6226R CPU @ 2.90GHz and NVIDIA RTX 3090 for every experiment.

## 4.2 Comparison with State-of-the-Art Methods

Tables 2 and 3 display the AUC and LOC measures for the five methods across the four datasets. The AUC measure is presented in Table 2 with varying the error margin $p$. RE-CURVE outperforms the other boundary detection methods, where the optimal representation approach varies for each dataset. RECURVE wins against TS-CP$^2$ in *all* datasets, irrespective of the evaluation measure. This finding demonstrates that the curvature is more effective for boundary detection in temporally coherent time series where the class changes gradually. WISDM, HAPT, and mHealth exhibit periodicity in certain classes, includ-

Table 2: Overall detection accuracy in the *AUC* measure (the best results **in bold**).

| Methods | $p$ | AUC ↑ | | | |
|---|---|---|---|---|---|
| | | WISDM | HAPT | mHealth | 50salads |
| RuLSIF | 5 | 0.559±0.005 | 0.797±0.001 | 0.598±0.002 | 0.606±0.006 |
| | 10 | 0.560±0.005 | 0.797±0.001 | 0.599±0.002 | 0.608±0.003 |
| | 20 | 0.563±0.005 | 0.797±0.001 | 0.600±0.002 | 0.611±0.004 |
| KL-CPD | 5 | 0.697±0.000 | 0.868±0.003 | 0.842±0.117 | 0.682±0.003 |
| | 10 | 0.702±0.000 | 0.873±0.004 | 0.849±0.113 | 0.684±0.003 |
| | 20 | 0.710±0.000 | 0.875±0.005 | 0.856±0.105 | 0.689±0.003 |
| TS-CP$^2$ | 5 | 0.815±0.012 | 0.692±0.007 | 0.560±0.014 | 0.680±0.010 |
| | 10 | 0.820±0.012 | 0.695±0.006 | 0.561±0.013 | 0.682±0.009 |
| | 20 | 0.823±0.013 | 0.697±0.006 | 0.561±0.010 | 0.685±0.008 |
| **RECURVE** **+TPC** | 5 | **0.897±0.003** | **0.909±0.001** | 0.954±0.003 | **0.719±0.005** |
| | 10 | **0.901±0.004** | **0.913±0.001** | 0.954±0.003 | **0.723±0.006** |
| | 20 | 0.902±0.003 | **0.919±0.001** | 0.955±0.005 | **0.729±0.006** |
| **RECURVE** **+TNC** | 5 | 0.880±0.004 | 0.863±0.017 | **0.979±0.004** | 0.594±0.016 |
| | 10 | 0.889±0.004 | 0.867±0.017 | **0.980±0.004** | 0.595±0.016 |
| | 20 | **0.905±0.004** | 0.876±0.017 | **0.980±0.005** | 0.600±0.015 |

ing `Walk` and `Run`. This periodicity would produce a closed shape for intra-segment trajectories and increase their curvatures in the representation space, enhancing the performance of RECURVE. In particular, when $p = 20$, RECURVE outperforms the second-best method by up to 12.7% in terms of the AUC measure for the mHealth dataset.

Table 3: Overall detection accuracy in the *LOC* measure (the best results **in bold**).

| Methods | LOC ↓ (thresholding by best F1) | | | | LOC ↓ (thresholding by mean segment length) | | | |
|---|---|---|---|---|---|---|---|---|
| | WISDM | HAPT | mHealth | 50salads | WISDM | HAPT | mHealth | 50salads |
| RuLSIF | 420.9±18.54 | 108.2±0.188 | 780.0±8.580 | 184.4±1.463 | 429.5±9.968 | 156.0±0.092 | 802.6±30.18 | 189.2±1.120 |
| KL-CPD | 189.0±12.20 | 121.5±4.540 | 306.4±126.5 | 179.5±3.853 | 198.3±2.329 | 113.0±2.545 | 352.6±119.7 | 176.6±1.017 |
| TS-CP$^2$ | 166.6±7.840 | 386.6±31.04 | 879.4±62.57 | 119.0±6.712 | 183.1±15.13 | 404.2±32.60 | 923.8±44.39 | 129.4±5.091 |
| **RECURVE+TPC** | **114.7±56.07** | **33.25±1.290** | 483.6±64.24 | **79.29±10.52** | **178.4±36.05** | **34.28±0.727** | 341.0±47.93 | **93.76±7.475** |
| **RECURVE+TNC** | 210.0±112.3 | 47.92±2.884 | **224.0±211.2** | 175.0±26.38 | 219.8±102.2 | 50.71±1.589 | **239.6±212.4** | 178.8±20.87 |

## 4.3 Detailed Investigation on Change Metric Quality

We display the average values of the change metrics separately for each pair of classes using the HAPT dataset, which was chosen for ease of visualization due to its small number of classes. Figure 6a depicts the *inter-class embedding distance*, which is determined by the Euclidean distance between the centroids of point representations of given classes. The values of the change metrics are averaged

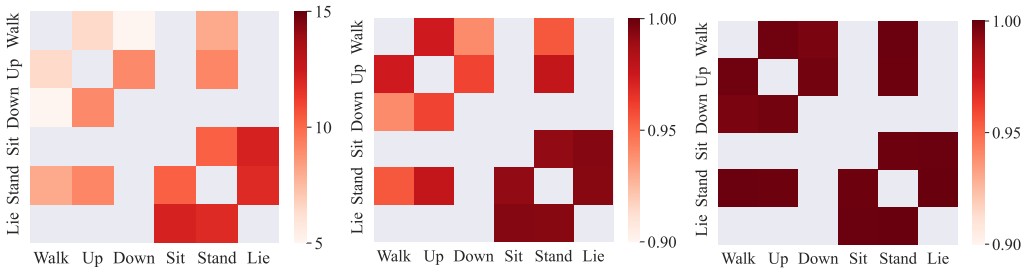

(a) Inter-class embedding distance.    (b) Averaged $\hat{y}_t^{\text{dist}}$ (TS-CP$^2$).    (c) Averaged $\hat{y}_t^{\text{curv}}$ (**RECURVE**).

Figure 6: Heatmaps of the inter-class distances and values of the change metrics between the classes in the HAPT dataset. A gray box indicates no transition between two classes.

across the *inter-segment* points for each distinct class transition. Figures 6b and 6c are obtained by the *distance-based* change metric $\hat{y}_t^{\text{dist}}$ of TS-CP$^2$ and the *curvature-based* change metric $\hat{y}_t^{\text{curv}}$ of RECURVE, respectively. Intriguingly, $\hat{y}_t^{\text{curv}}$ generates high values for *all* class pairs in Figure 6c, which indeed explains the overall high accuracy in Tables 2 and 3. In contrast, in Figure 6b, $\hat{y}_t^{\text{dist}}$ only generates high values when the inter-class embedding distance is sufficiently large (i.e., abrupt change), whereas it generates moderate values when the inter-class embedding distance is small (i.e., gradual change). That is, Figures 6a and 6b show a very high correlation. In summary, $\hat{y}_t^{\text{curv}}$ is insensitive to the degree of changes whereas $\hat{y}_t^{\text{dist}}$ is not. Therefore, this result demonstrates the superiority of the curvature-based change metric over the distance-based change metric.

Figure 7 magnifies six class pairs selected from all class pairs depicted in Figures 6b and 6c. For example, Stand→Sit and Lie→Stand are accompanied by rapid body movement, and both TS-CP$^2$ and RECURVE capture the boundaries well, as evidenced by the high density in the interval close to 1. In contrast, when two action classes are comparable, as in Stand→Walk, Down→Up, and Walk→Down, the values of the change metric of TS-CP$^2$ disperse to other intervals, resulting in a decrease in detection performance. RECURVE maintains the same shape in all density plots due to the remarkable effectiveness of our curvature-based change metric.

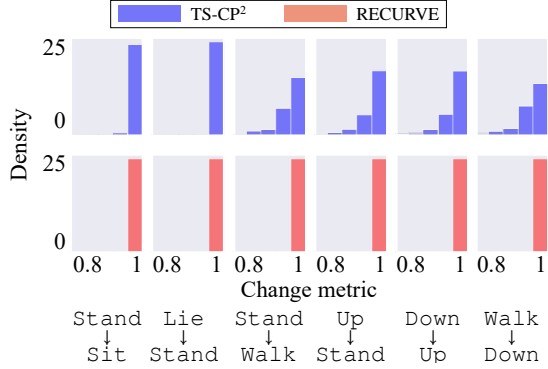

Figure 7: Distribution of the change metrics for each class transition in the HAPT dataset.

## 4.4 Evaluation Respective to Gradual and Abrupt Changes

Table 4 shows the corresponding AUC scores for gradual and abrupt changes in relation to various distance-based metrics. If the inter-class embedding distance between two classes is shorter than a certain threshold, the change between the two classes is categorized as *gradual*. Here, the threshold is established such that gradual changes represent 20% of all changes. Abrupt changes are excluded when measuring the improvement for gradual changes, and vice versa. The distance-based metrics, denoted by *DISTANCE*, are further categorized depending on whether the Euclidean distance or the cosine similarity is used. Note that RECURVE adopts the curvature-based metric.

Overall, RECURVE demonstrates greater improvements for gradual changes (AUC-Gradual) than for abrupt changes (AUC-Abrupt). For example, in the HAPT dataset with $p = 5$ and the TPC representation, the increase in the AUC measure from TS-CP$^2$ to RECURVE for gradual changes is 32%, which is significantly higher than the 21% increase for abrupt changes. This result confirms that the curvature-based method is particularly effective for detecting gradual changes because it captures subtle variations in the data that are not readily discernible through distance-based methods. In conclusion, RECURVE is versatile to support both gradual and abrupt changes.

Table 4: Detection accuracy in the AUC measure respective to gradual and abrupt changes.

| Methods | Repr. | $p$ | AUC-Gradual ↑ | | | | AUC-Abrupt ↑ | | | |
|---|---|---|---|---|---|---|---|---|---|---|
| | | | WISDM | HAPT | mHealth | 50salads | WISDM | HAPT | mHealth | 50salads |
| DISTANCE (Euclidean) | TPC | 5 | 0.690±0.008 | 0.516±0.029 | 0.519±0.005 | 0.596±0.005 | 0.694±0.009 | 0.716±0.010 | 0.696±0.010 | 0.622±0.027 |
| | | 10 | 0.690±0.008 | 0.520±0.029 | 0.521±0.005 | 0.599±0.005 | 0.695±0.009 | 0.717±0.010 | 0.699±0.010 | 0.624±0.027 |
| | | 20 | 0.691±0.009 | 0.527±0.028 | 0.521±0.005 | 0.603±0.005 | 0.696±0.008 | 0.718±0.009 | 0.700±0.010 | 0.628±0.027 |
| | TNC | 5 | 0.715±0.012 | 0.692±0.022 | 0.706±0.017 | 0.556±0.057 | 0.720±0.008 | 0.846±0.007 | 0.839±0.028 | 0.630±0.011 |
| | | 10 | 0.724±0.014 | 0.698±0.022 | 0.708±0.017 | 0.558±0.056 | 0.735±0.009 | 0.849±0.007 | 0.847±0.028 | 0.632±0.011 |
| | | 20 | 0.734±0.017 | 0.708±0.022 | 0.709±0.017 | 0.561±0.056 | 0.754±0.011 | 0.854±0.007 | 0.849±0.029 | 0.636±0.011 |
| DISTANCE (Cosine) | TPC (=TS-CP$^2$) | 5 | 0.807±0.014 | 0.602±0.023 | 0.546±0.015 | 0.671±0.015 | 0.838±0.009 | 0.746±0.021 | 0.631±0.011 | 0.703±0.039 |
| | | 10 | 0.811±0.014 | 0.606±0.023 | 0.548±0.014 | 0.674±0.010 | 0.845±0.010 | 0.747±0.021 | 0.636±0.011 | 0.706±0.035 |
| | | 20 | 0.814±0.015 | 0.613±0.022 | 0.550±0.011 | 0.677±0.009 | 0.847±0.010 | 0.747±0.020 | 0.637±0.010 | 0.714±0.037 |
| | TNC | 5 | 0.779±0.013 | 0.774±0.013 | 0.789±0.017 | 0.594±0.022 | 0.794±0.012 | **0.945±0.006** | 0.892±0.023 | 0.681±0.010 |
| | | 10 | 0.787±0.012 | 0.783±0.012 | 0.803±0.015 | 0.594±0.032 | 0.808±0.011 | 0.946±0.004 | 0.901±0.022 | 0.685±0.010 |
| | | 20 | 0.814±0.013 | 0.799±0.014 | 0.846±0.009 | 0.601±0.032 | 0.846±0.011 | 0.950±0.004 | 0.928±0.017 | 0.687±0.010 |
| **RECURVE** | TPC | 5 | **0.888±0.004** | **0.886±0.009** | 0.939±0.003 | **0.712±0.006** | **0.923±0.003** | **0.945±0.002** | 0.985±0.004 | **0.729±0.055** |
| | | 10 | **0.893±0.005** | **0.891±0.009** | 0.940±0.003 | **0.715±0.006** | **0.927±0.002** | 0.948±0.002 | 0.988±0.004 | **0.731±0.054** |
| | | 20 | **0.894±0.005** | **0.896±0.009** | 0.941±0.003 | **0.722±0.006** | **0.927±0.002** | 0.955±0.002 | 0.988±0.004 | **0.734±0.051** |
| | TNC | 5 | 0.867±0.007 | 0.773±0.038 | **0.975±0.003** | 0.551±0.034 | 0.901±0.006 | 0.931±0.002 | **0.988±0.006** | 0.600±0.016 |
| | | 10 | 0.875±0.007 | 0.779±0.038 | **0.977±0.003** | 0.553±0.034 | 0.910±0.006 | 0.933±0.002 | **0.989±0.006** | 0.602±0.016 |
| | | 20 | 0.890±0.008 | 0.791±0.038 | **0.977±0.004** | 0.558±0.033 | **0.927±0.006** | 0.939±0.002 | **0.990±0.006** | 0.606±0.015 |

## 4.5 Visual Analysis of the Change Metrics

Figure 8 visualizes the fluctuations of various change metrics obtained from the HAPT dataset using the TPC representation. TS-CP$^2$ fluctuates rapidly during the changes and seems to have many false negatives at the rightmost boundary area. However, RECURVE indicates the inter-segment points much more clearly than TS-CP$^2$, without excessive false positives and false negatives. The reliable detection is achievable by taking into account both the turning angle (the numerator in Eq. (5)) and the distance (the denominator in Eq. (5)) in the representation space.

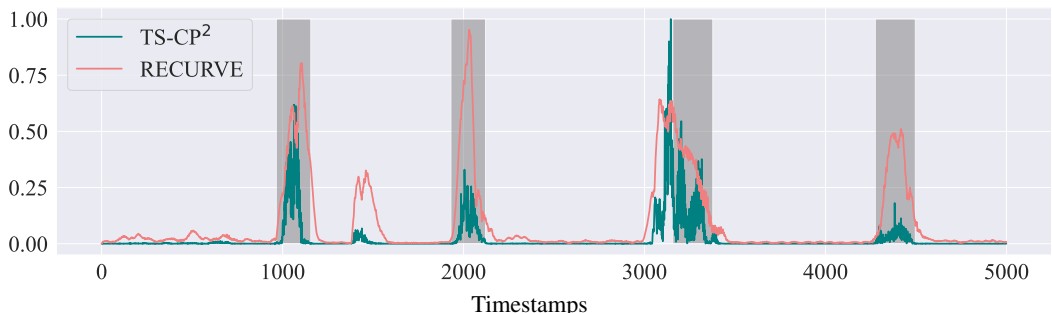

Figure 8: Change metric scores from the HAPT dataset with the default configuration. A gray-shaded area represents the inter-segment points between two class segments.

## 4.6 Sensitivity Analysis on the Hyperparameter *w*

Table 5 shows the performance of RECURVE while varying the hyperparameter $w$ (see Definition 3.3) when the error margin $p$ for the AUC measure is fixed at 10. The value of $w$ ranges from $0.25\times$ to $4.00\times$ of the default value, which is set to $5\%$ of the mean segment length (indicated by $1.00\times$). If the value of $w$ were too large, the denominator of Eq. (5) would be too large for any point in a time series, and the curvature would be unable to distinguish between intra- and inter-segment points. If the value of $w$ were too small, some noise in point representations would distort the curvature. Under this trade-off, the default value performs the best in terms of the AUC measure when it is averaged over the four datasets and the two representation learning methods. On a dataset with lengthy segments, such as mHealth, the sensitivity tends to decrease, and there is small variation when varying the value of $w$.

Table 5: Performance of RECURVE with varying the hyperparameter $w$ (the best results **in bold**).

| Repr. | $w$ | AUC ↑ | | | | LOC ↓ (thresholding by mean segment length) | | | |
|---|---|---|---|---|---|---|---|---|---|
| | | WISDM | HAPT | mHealth | 50salads | WISDM | HAPT | mHealth | 50salads |
| TPC | 0.25× | 0.832±0.015 | 0.901±0.004 | 0.911±0.008 | 0.685±0.007 | 358.7±89.01 | 40.41±2.233 | 654.7±36.04 | 136.9±2.064 |
| | 0.50× | 0.891±0.006 | **0.914±0.002** | 0.953±0.004 | 0.703±0.005 | 246.3±139.6 | 37.42±1.309 | 538.9±49.69 | 120.0±2.127 |
| | 1.00× | **0.901±0.004** | 0.913±0.001 | **0.954±0.003** | **0.723±0.006** | **178.4±36.05** | **34.28±0.727** | **341.0±47.93** | **93.76±7.475** |
| | 2.00× | 0.892±0.003 | 0.887±0.001 | 0.927±0.003 | 0.692±0.004 | 252.9±97.20 | 42.74±5.383 | 821.2±53.60 | 94.21±6.153 |
| | 4.00× | 0.861±0.002 | 0.847±0.002 | 0.893±0.004 | 0.604±0.004 | 273.8±119.0 | 53.00±10.77 | 628.2±39.36 | 104.0±3.502 |
| TNC | 0.25× | 0.824±0.016 | 0.842±0.011 | 0.956±0.009 | 0.580±0.015 | 249.7±37.59 | 52.95±3.698 | 213.7±110.2 | 222.0±7.443 |
| | 0.50× | 0.869±0.009 | 0.850±0.012 | 0.978±0.005 | 0.587±0.014 | 231.3±79.25 | 51.15±2.685 | **236.4±136.5** | 218.9±16.16 |
| | 1.00× | 0.889±0.004 | **0.867±0.017** | **0.980±0.004** | **0.595±0.016** | 219.8±102.2 | **50.71±1.589** | 239.6±212.4 | **178.8±20.87** |
| | 2.00× | **0.897±0.002** | 0.827±0.019 | 0.962±0.007 | 0.583±0.008 | **196.0±79.98** | 58.19±1.588 | 346.6±305.9 | 179.4±16.22 |
| | 4.00× | 0.871±0.002 | 0.773±0.016 | 0.937±0.007 | 0.568±0.009 | 265.5±58.16 | 97.38±6.328 | 265.6±92.20 | 183.2±17.64 |

The sensitivity analysis on the representation dimensionality $d'$ is available in Appendix C.

## 5 Conclusion

In this paper, we present RECURVE, a novel boundary detection method that uses the curvature of a representation trajectory to replace the consecutive distance for a change metric. Theoretically, the mean curvature of an intra-segment trajectory is greater than that of an inter-segment trajectory due to the confining nature of the representations of the points within a single class. Unlike the consecutive distance, this property of the curvature is insensitive to the degree of the changes between two classes (segments). Our comprehensive experiments confirm that RECURVE achieves up to 12.7% higher detection accuracy than state-of-the-art methods. Overall, we believe that our work pioneers a new direction for boundary detection in time series.

## Acknowledgments and Disclosure of Funding

This work was supported by Institute of Information & Communications Technology Planning & Evaluation (IITP) grant funded by the Korea government (MSIT) (No. 2020-0-00862 / RS-2020-II200862, DB4DL: High-Usability and Performance In-Memory Distributed DBMS for Deep Learning, 50% and No. 2022-0-00157 / RS-2022-II220157, Robust, Fair, Extensible Data-Centric Continual Learning, 50%).

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

# A  The Proof of Theorem 3.8 in a Special Case

To explain Theorem 3.8 in a more intuitive way, we assume a special case in the 2-dimensional space as shown in Figure 4. Let we have a random walk composed of three points $\mathbf{z}_{t-}$, $\mathbf{z}_t$, and $\mathbf{z}_{t+}$ in a 2-dimensional representation space. In the figure, there are three black dots in a row for the intra-segment trajectory (gray line) and the inter-segment trajectory (red line) respectively to visualize these three points in each trajectory. Here, $\mathbf{z}_t$ is an arbitrary point on the border of a confining circle $S_{C_i}$ or the enclosing circle. $\mathbf{z}_{t-}$ is the point on the line between the origin and $\mathbf{z}_t$. Finally, we consider $\mathbf{z}_{t+}$ as a random point distributed uniformly on the unit circle centered at $\mathbf{z}_t$ while the locations of $\mathbf{z}_t$ and $\mathbf{z}_{t-1}$ are fixed. In this case, we compare the average curvature at the single timestamp $t$ of the intra-segment trajectory and that of the inter-segment trajectory.

**Lemma A.1** (AVERAGE CURVATURE). *When the two points $\mathbf{z}_{t-}$ and $\mathbf{z}_t$ are fixed, the curvature $\kappa$ at $\mathbf{z}_t$ averaged over the location of $\mathbf{z}_{t+}$ is*

$$\kappa_{\mathcal{T}}(R) = \pi - \frac{1}{2}arccos\frac{1}{2R}. \tag{9}$$

*Proof.* When the confinement radius is $R$, the range of the turning angle is $[\pi - \arccos\frac{1}{2R}, \pi]$ using the cosine rule on the triangle composed of the points of intersection between $S_{C_i}$ and the unit circle centered at $\mathbf{z}_t$. $\qquad\square$

**Lemma A.2** (CHARACTERISTIC OF AVERAGE CURVATURE). *When $R > 1$, $\kappa_{\mathcal{T}}(R)$ decreases as the radius $R$ increases, i.e., $\kappa_{\mathcal{T}}(R)$ is a decreasing function of $R$.*

*Proof.* The derivative of $\kappa$ is $\frac{\mathrm{d}}{\mathrm{d}R}\kappa_{\mathcal{T}}(R) = -\frac{1}{2R\sqrt{4R^2-1}}$ and the derivative is always negative when $R > 1$. Therefore, $\kappa_{\mathcal{T}}(R)$ is a decreasing function of $R$. $\qquad\square$

Similar to Theorem 3.8, Proposition 3.5 and Lemma A.2 lead to Theorem A.3.

**Theorem A.3** (AVERAGE CURVATURE DIFFERENCE). *The average curvature of an intra-segment trajectory $\kappa_{\mathcal{T}_{intra}}(R_{intra})$ is greater than that of an inter-segment trajectory $\kappa_{\mathcal{T}_{inter}}(R_{inter})$.*

*Proof.* The radius of the confining circle of an inter-segment trajectory is always bigger than that of an intra-segment trajectory because an inter-segment trajectory traverses two distinctive class balls. As $\kappa_{\mathcal{T}}(R)$ is a decreasing function, $\kappa_{\mathcal{T}_{intra}}(R_{intra}) > \kappa_{\mathcal{T}_{inter}}(R_{inter})$. $\qquad\square$

Theorem A.3 concludes that the average curvature at $\mathbf{z}_t$ is greater if $\mathbf{z}_t$ is at the border of a smaller confining circle. This result can be extended to Theorem 3.8 intuitively. As $R$ increases, the probability of a trajectory hitting the border decreases as a random walk can traverse in a larger space. Therefore, the mean curvature decreases if a trajectory resides in a smaller confining circle.

# B  Hyperparameters for Compared Methods

For RuLSIF, we conduct a grid search for the learning rate (LR) $= \{0.05, 0.1, 0.2\}$, the weight of L2 normalization $\lambda_{L2} = \{0.01, 0.05, 0.1\}$, and the parameter of the RuLSIF loss $\alpha = \{0.01, 0.05, 0.1\}$. When applying RuLSIF to the four datasets, we use a multilayer perceptron with a single hidden layer with 100 units and train it with a batch size of 32 for 50 epochs. For KL-CPD, we conduct a grid search to determine the optimal hidden dimensionality $h = \{10, 50, 100\}$ of the RNN encoder/decoder, as well as the values for the hyperparameters $\lambda_{AE} = \{0.1, 0.01, 0.001\}$ and $\lambda_{Real} = \{0.1, 0.01, 0.001\}$ which govern the influence of the reconstruction loss and the MMD2 loss on real datasets. For training the generator of KL-CPD, the batch size is set to 64, the number of epochs is set to 3, and the learning rate is set to 0.001. Table 6 provides a summary of the determined hyperparameter values.

# C  Sensitivity Analysis on Representation Dimensionality

Table 7 shows the performance of RECURVE while varying the representation dimensionality $d'$ when the error margin $p$ for the AUC measure is fixed at 10. The value of $d'$ ranges from $0.25\times$ to $4.00\times$ of the default value, which is 8 for WISDM and HAPT or 32 for mHealth and 50salads

Table 6: Hyperparameter values of RuLSIF (left half) and KL-CPD (right half) after a grid search.

| Dataset | LR | $\lambda_{\text{L2}}$ | $\alpha$ | $\lambda_{\text{AE}}$ | $\lambda_{\text{Real}}$ | $\#hidden$ |
|---------|------|------|------|------|-------|-----|
| WISDM | 0.05 | 0.1 | 0.01 | 0.01 | 0.001 | 10 |
| HAPT | 0.2 | 0.01 | 0.01 | 0.01 | 0.1 | 10 |
| mHealth | 0.2 | 0.1 | 0.01 | 0.01 | 0.01 | 10 |
| 50salads | 0.05 | 0.01 | 0.05 | 0.1 | 0.01 | 50 |

(indicated by $1.00\times$). A trade-off point in the representation dimensionality exists for nearly all datasets. A representation space with an excessively high dimensionality is susceptible to the curse of dimensionality. If the value of $d'$ were too large, the turning angle and distance in Eq. (5) would be indistinguishable across all timestamps in a time series, as any two points in a high-dimensional space would become nearly orthogonal and their distance would always be similar. If the value of $w$ were too small, low-quality features would be extracted from the original time series by representation learning; thus, the performance degrades with an insufficient dimensionality as shown in the result of 50salads whose data dimensionality is $2048$. Overall, the default setting is suitable for achieving competitive performance for all datasets.

Table 7: Performance of RECURVE with varying the hyperparameter $d'$ (the best results **in bold**).

| Repr. | $d'$ | AUC ↑ | | | | LOC ↓ (thresholding by mean segment length) | | | |
|-------|------|-------|------|---------|---------|-------|------|---------|---------|
| | | WISDM | HAPT | mHealth | 50salads | WISDM | HAPT | mHealth | 50salads |
| TPC | $0.25\times$ | 0.870±0.007 | 0.844±0.011 | 0.942±0.007 | 0.719±0.006 | 349.9±32.99 | 307.0±30.03 | 553.4±69.70 | 97.64±6.153 |
| | $0.50\times$ | **0.906±0.003** | 0.836±0.168 | 0.949±0.006 | 0.719±0.007 | 377.5±443.7 | 104.4±130.3 | 586.1±39.86 | 97.33±5.553 |
| | $1.00\times$ | 0.901±0.004 | **0.913±0.001** | **0.954±0.003** | **0.723±0.006** | **178.4±36.05** | 34.28±0.727 | **341.0±47.93** | **93.76±7.475** |
| | $2.00\times$ | 0.882±0.017 | 0.905±0.003 | 0.942±0.005 | 0.718±0.005 | 180.6±59.32 | 35.62±2.669 | 600.9±36.82 | 100.9±3.814 |
| | $4.00\times$ | 0.857±0.015 | 0.900±0.004 | 0.937±0.006 | 0.719±0.007 | 200.4±55.87 | 38.27±1.191 | 592.3±64.23 | 100.0±6.432 |
| TNC | $0.25\times$ | 0.838±0.046 | 0.862±0.006 | 0.963±0.013 | 0.561±0.018 | 168.1±79.54 | 50.37±4.293 | 241.0±30.85 | 198.7±20.14 |
| | $0.50\times$ | 0.882±0.008 | 0.859±0.011 | 0.971±0.007 | 0.570±0.013 | **149.5±35.37** | **49.17±1.943** | 245.0±125.0 | 198.7±37.32 |
| | $1.00\times$ | **0.889±0.004** | 0.867±0.017 | **0.980±0.004** | 0.595±0.016 | 219.8±102.2 | 50.71±1.589 | **239.6±212.4** | **178.8±20.87** |
| | $2.00\times$ | 0.877±0.006 | 0.875±0.009 | 0.972±0.003 | 0.581±0.011 | 290.4±148.8 | 55.35±1.800 | 260.7±51.26 | 215.6±15.55 |
| | $4.00\times$ | 0.880±0.003 | **0.887±0.004** | 0.973±0.003 | **0.607±0.012** | 257.9±81.69 | 57.72±1.010 | 280.5±94.19 | 179.3±14.75 |

# D  Limitations

One notable limitation of RECURVE is the potential occurrence of false positives, particularly when dealing with short segment lengths that fall below the predefined threshold, denoted as $w$. For instance, consider a scenario where there is a rapid transition from one activity class to another, such as transitioning from walking for an extended period to a brief sprint, followed by resuming walking. In such cases, if the segment length is shorter than the specified $w$ duration, every timestamp within the brief sprint segment might erroneously be identified as a boundary, leading to a number of false positives. To mitigate this issue, we have implemented a strategy where $w$ is determined as 5% of the mean segment length. However, it is worth noting that this approach may encounter challenges in rare cases where the segment lengths exhibit significant variance. We will further investigate how segment length variance impacts the efficacy of RECURVE as future work.

