# OpenReview forum: "Exploiting Representation Curvature for Boundary Detection in Time Series"
_NeurIPS.cc/2024/Conference — NeurIPS 2024 poster_

### Official Review · Reviewer_PiYP · 2024-07-05

**Soundness:** 3
**Presentation:** 4
**Contribution:** 2
**Rating:** 5
**Confidence:** 4

**Summary:**

This paper proposes a novel approach called RECURVE (Representation trajEctory CURVaturE) for boundary/change point detection based on time series representations. RECURVE measures changes in representations of time series windows over time, based on curvatures instead of distances in the representation space like existing work. The intuition is that the direction of the representation trajectory tends to change more sharply at points within a segment than at points between segments. This is because representation learning aims to learn class-separated features, and intra-segment points are confined within a class-specific ball, while inter-segment points are not. RECURVE works by first deriving a representation trajectory using a time-series representation learning method, then calculating the curvature at each point to identifying boundaries.

**Strengths:**

$\textbf{Presentation: }$ The paper is well-written and easy to follow. The graphics are informative and the evaluation has an extensive set of tests.

$\textbf{Problem definition: }$ The problem this paper is tackling is an important one. Identifying states and state changes in time series is very important for many application. Most change point detection methods are designed to capture abrupt changes in data distribution which is not inline with the reality of state transitions in real-world data. The proposed method claims to capture such gradual transitions better than these existing methods.

**Weaknesses:**

$\textbf{Class separated representations: }$ The theoretical analysis relies on Definition 3.4 which states that contrastive-based representations learning produces class-separated representations. First, there is no guarantee of class separation for contrastive-based methods, and in many complex datasets this doesn't in fact happen. If the proposed method is only justified under class-separated settings, this should be clearly stated and the authors should propose a way to assess if the representations generated by a contrastive framework qualify for this. Second, In the case of perfect class separation, a clustering method can be used to identify the underlying states for all windows and therefore find the time of transition between the states. Can the authors justify their reasoning regarding this issue?

$\textbf{Normalizing the change metric: }$ A limitation of many change point detection (CPD) methods is that they normalize the change metric in order to obtain a normalized score. This means the largest change in a single time series sample will always be assigned a CPD score of 1, even if the time series sample has no true change in state. This also implies that the value of the CPD scores are not comparable across samples because they are normalized per sample. This limitation applies to the proposed method as well and the authors need to elaborate on this issue more. Aside from that, this problem will also have implication on the choice of threshold used to identify the change points.

$\textbf{Baselines: }$ An important baseline that is missing from the evaluations is a distance-based change point detection model that uses the same representation learning framework as RECURVE. The main claim of the paper is that the curvature is a better measure of change, so this needs to be evaluated in the evaluation section. I would suggest having baselines similar to RECURVE+TNC and RECURVE+TPC, but only using a similarity metric like cosine similarity or even euclidian distance (DISTANCE+TNC and DISTANCE+TPC). Such baseline can also show the advantage of the proposed method in samples with gradual transition.

**Questions:**

$\textbf{Change metric quality: }$ In section 4.3, the authors show that the change metric of RECURVE is on average always high for transition points while distance-based metrics have varying score values depending on the state change. My question is why having varying degrees is a bad thing? Because in many cases, this can also show us the similarity or difference between the two transition states (Walking to sitting vs walking to lying down).

$\textbf{Scores over time: }$ Please include a figure that shows the estimated CPD scores for a sample over time. This will be very helpful to understand the behaviour of the method, in particular around the change point boundary.

$\textbf{Error margin: }$ The choice of error margin seems to be an important one, especially to show how the proposed method behaves in slow-transition states. Can you provide performance results across multiple error margin values in the appendix and discuss your observation.

**Limitations:**

Overall, the limitations are well-covered in the paper. Some points that I mentioned in earlier sections can be added to this discussion.

---

> ### Author Rebuttal · Authors · 2024-08-06
>
> **Thank you so much for acknowledging the impact and intuitiveness of our new boundary detection method.  We hope that our responses have addressed your concerns and that you are able to raise your rating.**
>
> ---
>
> **`W1`** *Theoretical analysis with class-separated representations.*
>
> **(1)** You are right. There is no guarantee of class separation, **even though it tends to hold in realistic settings [18, 19]**. As you might agree, theoretical formulation often needs some kind of assumptions on data distribution. Without the class confinement assumption, RECURVE is empirically proven to work well with real-world datasets, because point representations are at least densely populated for intra-segment points and sparsely populated for inter-segment points, under the reasonable quality of the learned representations. Figures 5, 6, and 7 show that the curvature-based scores are higher for inter-segment points than for intra-segment points. In particular, the 2-dimensional plots in Figure 5 show the class-separated representations.
>
> **(2)** In fact, we did try the clustering approach. Our experience indicates that there is no guarantee of one-to-one correspondence between clusters and classes. For example, under the "Walk" class, multiple clusters are formed depending on the speed of walking. Thus, clustering (i.e., proper setting of the number of clusters) was not that straightforward for this purpose.
>
> ---
>
> **`W2`** *Normalization of the change metric.*
>
> Because change point detection (CPD) methods are typically applied to **long, continuing time series** (e.g., smartphone sensor recordings for a full day), it is reasonable to think that there are state changes. **The same normalization is applied to *all* samples from an individual dataset by treating them as a single time series**. Thus, we believe that all your concerns on the normalization should be resolved now. The details on the normalization will be clarified in the final draft. Similarly, the threshold $\varphi$ is set for each individual dataset, as described in Section 3.3.
>
> ---
>
> **`W3`** *Missing baselines.*
>
> Thank you for your comment. We already have included such baseline for comparison. In fact, what you refer to as DISTANCE(cosine)+TPC is TS-CP$^2$. Per your comment, we add similar baselines, DISTANCE(cosine)+TNC, DISTANCE(Euclidean)+TPC, and DISTANCE(Euclidean)+TNC. Please refer to Table 5 of the PDF file [🔗](https://openreview.net/attachment?id=XDnlT4Yx3m&name=pdf). Again, **RECURVE is shown to outperform all such baselines**. Specifically, in the HAPT dataset with $p=5$, the increase in the AUC from TS-CP$^2$ to RECURVE+TPC for gradual changes is 32\%, which is significantly higher than the 21\% increase for abrupt changes. This finding indicates that RECURVE is notably more efficient in handling gradual changes.
>
> ---
>
> **`Q1`** *Quality of the change metric.*
>
> This is a really good question! As long as the score values of inter-segment points are mostly higher than those of intra-segment points, having varying degrees is not a bad thing, as you thought. However, as shown in the left plot of Figure 1, a wide range of the score values of inter-segment points may overlap with the score values of intra-segment points, especially for gradual changes. Thus, **the accuracy of change point detection is degraded by this overlap in the distance-based method**.
>
> ---
>
> **`Q2`** *Figure for the scores over time.*
>
> Absolutely. Please see Figure 8 of the PDF file [🔗](https://openreview.net/attachment?id=XDnlT4Yx3m&name=pdf). **The CPD scores of RECURVE indicate the inter-segment points much more clearly than those of TS-CP$^2$**.
>
> ---
>
> **`Q3`** *Performance results across multiple error margins.*
>
> Table 2 already contains the performance results across multiple error margins. Per your suggestion, **we present the performance results separately for gradual and abrupt changes across multiple error margins in Table 5** (see the PDF file [🔗](https://openreview.net/attachment?id=XDnlT4Yx3m&name=pdf)). As the error margin $p$ increases, the AUC metric increases slightly more for gradual changes than for abrupt changes, though the difference is not very clear. We conjecture that the transition strength is a significant factor in distinguishing between gradual and abrupt changes, rather than the transition duration. Table 5 will be added to the final draft.

---

> > ### Comment · Reviewer_PiYP · 2024-08-11
> > **Rebuttal response**
> >
> > This rebuttal has been very helpful to clarify my main concerns. I believe the additional results and figures in the appended pdf will be important additions to the paper.
> > My first concern about class-separation still exists. I think the authors should definitely discuss this in the paper because it will be an important thing to consider if someone wants to pick a representation learning framework to combine with the proposed method.
> > But overall, I'm happy to increase my score based on this rebuttal.

---

> > > ### Author Response · Authors · 2024-08-12
> > > **Thank you for your feedback**
> > >
> > > We are pleased that you find our responses to be satisfactory. Thank you again for your valuable and insightful feedback. We will clearly discuss the issues about class-separated representations and surely add the additional results to the final version.

---

### Official Review · Reviewer_aPkZ · 2024-07-10

**Soundness:** 3
**Presentation:** 3
**Contribution:** 3
**Rating:** 7
**Confidence:** 4

**Summary:**

The authors propose a novel boundary detection method for time series based on measuring the curvature of the local trajectory of a learned per-timepoint embedding/representation. Some theoretical justifications are provided for the proposed method. Empirically, it is shown to work better than a number of existing methods on a few time series datasets.

**Strengths:**

The proposed method seems novel and the empirical results are encouraging. The paper overall is easy to read and the main idea is easy to understand.

**Weaknesses:**

The whole idea builds on the fact that there exists a learned representation that more or less satisfies the locality assumption (e.g. intra-class points live within a hypersphere). Although some limited empirical evidence is presented, it is unclear in general whether this is a common phenomenon or something else needs to be done to encourage representations that more faithfully respect the assumption.

**Questions:**

I wonder whether one can go a step further and establish some notion of optimality under the right settings. Suppose that the representation for each class is drawn from a gaussian with class-dependent mean/covariance, then there exist optimal statistical tests for the inter-class change points. How would the proposed method perform compared to the optimal boundary detector?

---

> ### Author Rebuttal · Authors · 2024-08-06
>
> **Thank you so much for acknowledging the novelty, evaluation result, and intuitiveness of our new boundary detection method.**
>
> ---
>
> **`W1`** *Locality assumption on the learned representations.*
>
> Thank you for your insightful comment. Yes, RECURVE is built upon class-separated representations. **We believe that it is a widely known fact that representation (contrastive) learning produces class-separated representations [18, 19]**. According to [18], the augmented positive examples form a connected graph based on augmentation overlap; thus, the contrastive learning will cluster the examples of the same class together and lead to class-separated representations. We use two representation learning techniques, TPC and TNC, for our evaluation. Consecutive classes tend to be well separated on the embedding space, as shown in Figure 5. RECURVE is empirically shown to work well with these two representation learning techniques.
>
> As a topic for future work, we will continue to explore the design of a customized representation learning technique for RECURVE. Your comment is indeed inspiring!
>
> [18] Chaos is a Ladder: A New Theoretical Understanding of Contrastive Learning via Augmentation Overlap, NeurIPS 2022
>
> [19] InfoNCE Loss Provably Learns Cluster-Preserving Representations, COLT 2023
>
> ---
>
> **`Q1`** *Optimality under the right settings.*
>
> This is a very good suggestion! The Gaussian mixture model can be applied to a set of point representations. Then, based on the probability distribution of a point, some statistical tests can be done to distinguish between intra-segment points and inter-segment points. It is difficult to implement this procedure during the rebuttal period because of the short time allotted, so we will include this comparison in the final draft. Given this setting, where point representations are well clustered, we anticipate that RECURVE will be very close to the optimal boundary detector.

---

> > ### Comment · Reviewer_aPkZ · 2024-08-13
> >
> > Thanks for the response, I'll keep my score.

---

> > > ### Author Response · Authors · 2024-08-13
> > > **Thank you for your feedback**
> > >
> > > We greatly appreciate your valuable feedback and strong support.

---

### Official Review · Reviewer_vKgw · 2024-07-10

**Soundness:** 3
**Presentation:** 3
**Contribution:** 3
**Rating:** 6
**Confidence:** 4

**Summary:**

This paper proposed to use curvature as the metric to detect the boundaries. Empirical and experimental results also show that the confining box is different for inter- and intra- variable data points.

**Strengths:**

- The idea of using curvature for boundary detection is novel in time series forecasting.
- The paper is well-written, figures are clear and informative, making the paper easy to follow.

**Weaknesses:**

- The paper lacks a strong foundational motivation for introducing the curvature-based boundary detection method. The rationale behind why curvature would be a better metric compared to existing methods is not convincingly argued. From Figure 5, it seems that the distance-based methods can also split (a) and (c) well while for (b), the curvature difference is also not very big.
- Many related boundary detection methods are mentioned in the related work but not directly compared in the experiment. It would be beneficial to include a more comprehensive comparison to show the effectiveness of the proposed approach.
- RECURVE is highly dependent on the quality of the learned representation which makes it also holds the same limitation as a distance-based model when the representation of a time series data is not properly learned.

**Questions:**

- How is the sensitivity of the parameter analysed? Are the results presented with the test set or is it with the evaluation set?
- In Figure 5 (b), the curvature of some continuous intra-variable data points seems to have a similar curvature (approximately a straight line) compared to the curvature between the variables from a different class. Can you clarify this observation and how it may affect the proposed method?
- What is the computational cost of the proposed method compared to baselines?

**Limitations:**

The limitation is well stated in Section D.

---

> ### Author Rebuttal · Authors · 2024-08-06
>
> **Thank you so much for acknowledging the novelty and intuitiveness of our new boundary detection method.**
>
> ---
>
> **`W1`**, **`Q2`** *Lack of a strong foundational motivation and interpretation of Figure 5.*
>
> **(1)** We anticipate that our clarification will effectively address your concerns.
> * ***Rationale***: In Introduction, Figure 1 empirically shows that the distances between consecutive points largely overlap between intra-segment points and inter-segment points, especially for gradual changes (Up$\leftrightarrow$Down). Figure 2 illustrates the foundational motivation of the curvature-based metric. As long as the point representations are confined into a class ball, the trajectory of intra-segment points make sharp turns more frequently than that of inter-segment points, regardless of gradual and abrupt changes. This benefit of the curvature is proven in Theorem 3.8.
> * ***Evaluation***: Furthermore, **Figures 6 and 7 clearly demonstrate the advantage of the curvature-based metric over the distance-based metric**. For gradual changes, e.g., from walking to descending (Walk$\rightarrow$Down), the embedding distances across these two classes are small (Figure 6(a)). Consequently, the distance-based scores become low for these transitions because the embedding distances between consecutive inter-segment points should be also small (Figure 6(b) and the top row of Figure 7). In contrast, regardless of gradual and abrupt changes, the curvature-based scores are sufficiently high (Figure 6\(c) and the bottom row of Figure 7).
>
> **(2)** **Figure 5 is obtained using two principle components out of 32 dimensions**. Thus, the exact position of each point representation may not be reflected on the 2-dimensional plot. The color of a point indicates the exact curvature score calculated in the 32-dimensional space. The purpose of Figure 5 is to support our motivation that the intra-segment points are somewhat clustered whereas the inter-segment points are not. In order to minimize confusion, we will identify the plots that most accurately represent the original positions in the 32-dimensional space.
>
> In conclusion, we trust that the curvature-based metric's superiority over the distance-based metric is readily apparent to you. We will endeavor to further improve the presentation in the final draft.
>
> ---
>
> **`W2`** *Comparison with more baseline methods.*
>
> In fact, we choose the representative method from each of the three categories (statistics-based, pattern-based, and representation-based) mentioned in Related Work. Because the representation-based methods are regarded as the state of the art, we add another representation-based method, which is a variation of TS-CP$^2$ with the TNC representation learning technique. **RECURVE also beats this additional method**, and please see Table 5 in the PDF file [🔗](https://openreview.net/attachment?id=XDnlT4Yx3m&name=pdf).
>
> ---
>
> **`W3`** *Dependency on the learned representation.*
>
> We agree with you. However, as representation learning (or contrastive learning) for time series advances remarkably [a], it is unlikely that the quality of the learned representation is very poor. We will discuss this limitation in Appendix D of the final draft. It is crucial to note that, **even if the quality of the learned representation is sufficiently high, the distance-based method is susceptible to gradual changes whereas the curvature-based method is not**.
>
> [a] Universal Time-Series Representation Learning: A Survey, arXiv:2401.03717, 2024
>
> ---
>
> **`Q1`** *Hyperparameter sensitivity analysis.*
>
> We have conducted an analysis of the sensitivity of two hyperparameters $w$ and $d^\prime$, as shown in Section 4.4 and Appendix C, respectively. The sensitivity analysis was done using the test set. It was confirmed that the default setting of the two hyperparameters is suitable for achieving competitive performance across for all datasets, and the sensitivity was not a concern.
>
> ---
>
> **`Q3`** *Computational cost.*
>
> Given a representation dimensionality $d^\prime$ and a time-series length $T$, the computational complexity of curvature computation in RECURVE is $O(d^{\prime}T)$. **This complexity is the same as that of TS-CP$^2$, thereby ensuring a comparable computational burden**. It is noteworthy that with a low representation dimensionality, i.e., $d^\prime=32$ as the default, and a linear scaling with respect to the length of the time series, RECURVE demonstrates efficiency in the context of lengthy time series. The detailed cost analysis will be added in the final draft.

---

> > ### Comment · Reviewer_vKgw · 2024-08-12
> >
> > The authors have provided clear and informative responses to my concerns.
> >
> > I maintain my positive score on this paper.

---

> > > ### Author Response · Authors · 2024-08-13
> > > **Thank you for your feedback**
> > >
> > > We are happy to hear that our responses are satisfactory to you. Thank you again for your valuable comments and suggestions. We will carefully incorporate your comments into the final version.

---

### Official Review · Reviewer_ZR9k · 2024-07-12

**Soundness:** 3
**Presentation:** 3
**Contribution:** 3
**Rating:** 7
**Confidence:** 1

**Summary:**

This paper proposes a novel boundary detection method based on the curvature of a representation trajectory. The feasibility of the proposed algorithm is analyzed intuitively and theoretically, and the proposed method is experimentally shown to have good performance.

**Strengths:**

1. The proposed method is simple, but effective.
2. The proposed method is shown to be effective intuitively, theoretically and experimentally.

**Weaknesses:**

It may be better to first classify the boundaries into several categories, and then discuss the related work and experimental comparison for these categories respectively.

**Questions:**

1. How many types of boundaries are there?
2. Which boundary problems can the proposed method solve and which boundary problems cannot it solve?

**Limitations:**

Yes

---

> ### Author Rebuttal · Authors · 2024-08-06
>
> **Thank you so much for acknowledging the effectiveness and intuitiveness of our new boundary detection method. We hope that you can support our work during the discussion period.**
>
> ---
>
> **`W1`**, **`Q1`** *Categorization of the boundaries and evaluation with the categorization.*
>
> **(1) *Categorization***: Thank you for your valuable comment. In our current draft, we categorize boundaries into **gradual** and **abrupt** changes depending on the speed of change. Gradual changes (e.g., Walk$\rightarrow$Down) involve slow transitions over time, which can be challenging for traditional distance-based metrics to detect. Abrupt changes (e.g., Stand$\rightarrow$Sit), on the other hand, involve rapid transitions that are typically easier to identify due to significant differences between consecutive points.
>
> One way to explicitly categorize the boundaries is to use the inter-class embedding distance, which is the Euclidean distance between the centroids of point representations of given classes. If the distance is below a certain threshold, the transition between the two classes is considered gradual; otherwise, it is considered abrupt.
>
> **(2) *Evaluation***: Using Figures 6 and 7 in Section 4.3, we have already analyzed our experimental results with respect to the categorization. This analysis enables us to demonstrate how our method, RECURVE, performs in different scenarios and highlights its effectiveness in detecting both types of changes. To further improve the structure and clarity of our paper, we will expand Related Work to include more references on gradual change point detection.
>
> ---
>
> **`Q2`** *Capability of RECURVE according to the categorization.*
>
> Per your suggestion, we show the performance results separately for gradual and abrupt changes in Table 5 (see the PDF file [🔗](https://openreview.net/attachment?id=XDnlT4Yx3m&name=pdf)). It is obvious that **RECURVE can support both types of boundaries**, also as demonstrated in Section 4.3. Specifically, in the HAPT dataset with $p=5$, the increase in the AUC from TS-CP$^2$ to RECURVE+TPC for gradual changes is 32\%, which is significantly higher than the 21\% increase for abrupt changes. The findings presented in Table 5 indicate that RECURVE is notably more efficient in handling gradual changes. A tricky case with very short segments is discussed in Appendix D.

---

> ### Comment · Reviewer_ZR9k · 2024-08-13
>
> Thanks for your responses, I will keep my positive score.

---

> ### Author Response · Authors · 2024-08-13
> **Thank you for your feedback**
>
> We greatly appreciate your valuable feedback and encouragement. Your comments will be definitely incorporated into our final version.

---

### Official Review · Reviewer_EoQb · 2024-07-13

**Soundness:** 3
**Presentation:** 3
**Contribution:** 3
**Rating:** 7
**Confidence:** 4

**Summary:**

The paper introduces a boundary detection method called RECURVE for time series data, which leverages the curvature of representation trajectories as a novel change metric to accommodate both gradual and abrupt changes. Experiments on diverse real-world datasets demonstrate that RECURVE outperforms state-of-the-art methods, achieving up to a 12.7% improvement in detection accuracy.

**Strengths:**

1. Innovation. The paper introduces an innovative boundary detection method that identifies boundary points by analyzing the curvature of representation trajectories in time series data, offering a new perspective to the field.

2. Intuitiveness: The authors intuitively demonstrate their points through Figures 1 and 2, which serve to elucidate the authors' insights and are persuasive.

3. Theoretical Support: The paper substantiate the correctness of the proposed viewpoint through rigorous theoretical analysis, providing a solid theoretical foundation for the method.

4. Experimental Validation: Extensive experimental design and results indicate that the proposed new metric is effective and well supports the paper's claims.

5. Writing Quality: The writing is clear, logically coherent, and easy for readers to understand.

**Weaknesses:**

1. While the concept introduced in this paper is highly innovative—it detects boundary points from the perspective of representation trajectories— the intrinsic advantages of using a curvature metric have not been fully articulated:

- 1.1 Although the paper proposes a method based on the trajectory perspective, it appears to rely primarily on data from three points in the trajectory (t+1, t, t-1), which leads to a final metric design that is similar to distance-based methods.

- 1.2 Looking at Equation 3.2, there is a certain correlation between the curvature-based method and the distance-based method: k_t is negatively correlated with distance. This raises the question of whether the curvature-based method might actually subsume the distance-based method. Is the introduction of the turning angle theta truly necessary? The authors should delve into what the RECURVE method solves that distance-based methods cannot.

- 1.3 Why is the curvature of intra-class points so large? Is this due to an excessively large theta or a too-short distance? Similarly, why is the curvature of inter-class points so small? Is this due to a too-small theta or a too-long distance?

2. Assumption 3.6 raises some concerns. In this assumption, the authors posit that ||z_t − z_{t−1}|| equals 1. However, this assumption appears to be difficult to uphold in boundary point detection tasks; otherwise, distance-based methods should be universally ineffective. If this assumption does not hold, what are the implications for the perspectives expressed in this paper?

3. Minor Question: The definition of Mean Total Curvature is somewhat perplexing; could you clarify why it is termed "Mean Total" rather than simply "Mean"?

4. Other Inquiry: I have some queries regarding the Case Study. In Figure 5(c), I have manually marked five points: a, b, c, d, and e (you can view this through [this link](https://anonymous.4open.science/r/temp-AC2C/5c.png)). The curvature at point b, denoted as k_b=\theta_b/(ab+bc), and at point d, k_d=\theta_d/(cd+de). Clearly, \theta_b is an obtuse angle and \theta_d is an acute angle, implying \theta_b > \theta_d. Additionally, visually assessing the distances on the image suggests that (ab+bc)<(cd+de). Consequently, it would be expected that k_b>k_d. Considering that a smaller k indicates an inter-segment point, why does Figure 5(c) depicts b as an inter-segment point and d as an intra-segment point. Am I misunderstanding something here?

**Questions:**

See weakness. I'd be happy to raise my score if my questions were answered well.

**Limitations:**

Yes.

---

> ### Author Rebuttal · Authors · 2024-08-06
>
> **Thank you so much for acknowledging the innovation of our new boundary detection method. We really hope that our responses are satisfactory to you.**
>
> ---
>
> **`W1.1`** *Similarity to the distance-based metrics.*
>
> The three points used for calculating the curvature are $t-w$, $t$, and $t+w$. Here, $w$ was introduced for stability. Then, the **simplified trajectory** (i.e., line segment) between $t-w$ and $t$ and that between $t$ and $t+w$ are considered to measure the turning angle at $t$. Thus, we believe that the trajectory perspective is incorporated into the design of our metric.
>
> ---
>
> **`W1.2`** *Unclear necessity of the turning angle.*
>
> Thank you very much for your insightful comment. Yes, you are right. The curvature seeks to maximize the **combined effect** of the turning angle $\theta$ and the distance. In fact, Definition 3.2 is similar to the conventional curvature estimation in the geometry field [17]. **The turning angle is truly necessary especially for gradual changes.** Please see Figures 6 and 7. Between similar actions, e.g., from walking to descending (Walk$\rightarrow$Down), the embedding distances across these two classes are small (Figure 6(a)). Consequently, the distance-based scores become low for these transitions because the embedding distances between consecutive inter-segment points should be also small (Figure 6(b) and the top row of Figure 7). Nevertheless, these drawbacks are fully cured by adding the turning angle to the change metric (Figure 6\(c) and the bottom row of Figure 7). **Overall, Figures 6 and 7 clearly show how crucial the turning angle is to the change metric.**
>
> [17] Curvature and torsion estimators based on parametric curve fitting. Computers & Graphics, 29(5):641–655, 2005.
>
> ---
>
> **`W1.3`** *Roles of the turning angle and the distance.*
>
> The relative contribution of the turning angle $\theta$ and the distance varies depending on the scenario. For gradual changes such as the transition from walking to descending in Figure 6, the score increases primarily due to a relatively large $\theta$. In contrast, for abrupt changes such as the shift from walking to standing, the distance term has a greater impact on the score. In order to substantiate our assertion, we present the values of $\theta$ alone in **Figure 9(d)** (refer to the PDF file [🔗](https://openreview.net/attachment?id=XDnlT4Yx3m&name=pdf)). Here, **$\theta$ is greater for the transitions where the curvature-based method succeeds but the distance-based method fails**---i.e., for the transitions with smaller inter-class embedding distances.
>
> ---
>
> **`W2`** *Implication of Assumption 3.6.*
>
> This assumption is made in order to **concentrate on the effect of the turning angle especially for gradual changes**, where the distances between consecutive points are similar within a **local** range due to the temporal coherence of time series. This intention will be further elucidated in the final draft. That is, Theorem 3.8 is presented to prove that, **because of the effect of the turning angle**, the curvature is larger for intra-segment points than for inter-segment points.
>
> We acknowledge that the assumption is not always true for datasets from the real world. However, RECURVE continues to be reliable because it takes advantage of the combined effect of the turning angle and the distance. Moreover, we **empirically** confirm the validity of Theorem 3.8 on the real-world datasets **without the assumption**, as shown in Figure 7.
>
> ---
>
> **`W3`** *Minor: Term "mean total curvature".*
>
> Sorry about the perplexity in the term. We will surely correct the term as the *mean curvature*.
>
> ---
>
> **`W4`** *Other Inquiry: Actual calculation of the curvature in Figure 5.*
>
> We are really impressed by your thorough review. Definition 3.2 and Line 165 of our draft state that points spaced $w$ apart are used in the curvature calculation rather than just consecutive points. This calculation makes the curvature insensitive to local noise and enables us to capture broader changes in the trajectory. Your understanding is correct except this point. The discrepancy between actual curvature scores and your calculations is mainly due to dimensionality reduction, where a 32-dimensional representation space is converted to a 2-dimensional space. Thus, unfortunately, the precise location of each point representation may not be accurately depicted on Figure 5.
>
> Additionally, we have removed some redundant points from each trajectory visualization, showing only one point out of every ten, to prevent the figure from becoming too crowded and to make the overall trajectory easier to understand. Our visualization may have resulted in some confusion, and we will surely try to eliminate this confusion.

---

> > ### Comment · Reviewer_EoQb · 2024-08-09
> >
> > Thanks for the insightful reply. The authors have solved most of my concerns, and I have raised my score.

---

> > > ### Author Response · Authors · 2024-08-10
> > > **Thank you for your feedback**
> > >
> > > We are pleased that you find our responses to be satisfactory.  Thank you again for your valuable and insightful feedback.  We will incorporate your comments into the final version.

---

### Author Rebuttal · Authors · 2024-08-06

We deeply appreciate your considerate feedback on our paper. Overall, **we are delighted that most of the reviewers agreed with three main contributions**: **(1) novelty** &mdash; "the idea of leveraging the curvature of a representation trajectory is novel and innovative" (Reviewers EoQb, ZR9k, vKgw, and aPkZ); **(2) effectiveness** &mdash; "RECURVE outperforms state-of-the-art methods" on diverse real-world datasets (Reviewers EoQb, ZR9k, vKgw, and aPkZ); **(3) thoroughness** &mdash; "the feasibility of the proposed algorithm is analyzed intuitively and theoretically" (Reviewers EoQb, ZR9k, and aPkZ). To further clarify our contributions, we have addressed the issues raised as follows.

* **Assuring necessity of the curvature** (Reviewers EoQb, ZR9k, vKgw, and PiYP): We have elaborated the claim that the curvature is essential for handling gradual change points which distance-based metrics cannot capture. Moreover, we support this claim with Table 5 and Figure 9 in the PDF file [🔗](https://openreview.net/attachment?id=XDnlT4Yx3m&name=pdf).
* **Elucidating theoretical assumptions** (Reviewers EoQb, vKgw, aPkZ, and PiYP): We explain the rationale behind assumptions and justify each assumption using recent literature.
* **Enriching experiment results** (Reviewers ZR9k, vKgw, and PiYP): To further demonstrate RECURVE's superiority over other change point detection methods, we have added three more baselines and evaluated each method with respect to the category of change points (refer to Table 5 in the PDF file [🔗](https://openreview.net/attachment?id=XDnlT4Yx3m&name=pdf)).
* **Clarifying visualization** (Reviewers EoQb, vKgw, and PiYP): We eliminate confusions in Figure 5 by describing the configurations for visualizing the trajectories. We also visualize change metric values over time in Figure 8 of the PDF file [🔗](https://openreview.net/attachment?id=XDnlT4Yx3m&name=pdf).

We sincerely hope that our responses have satisfied the reviewers. Also, we welcome further discussions throughout the discussion period. Thank you again for your time and effort!

---

### Decision · Program_Chairs · 2024-09-25

**Decision:**

Accept (poster)

**Comment:**

This paper introduces a novel method, RECURVE, which utilizes the curvature of representation trajectories to detect boundaries in time series data. After a comprehensive evaluation, I recommend accepting this paper for the following reasons. The primary strength of this paper lies in its innovative approach to boundary detection. By analyzing the curvature of representation trajectories rather than relying solely on distance-based metrics, the method effectively handles both gradual and abrupt changes in time series data. This approach addresses a critical gap in existing methods, which often struggle with gradual transitions. The authors provide strong theoretical foundations for their approach, supported by rigorous empirical validation across diverse real-world datasets, where RECURVE consistently outperforms state-of-the-art methods. The paper is also well-structured and clearly written, making it accessible to a broad audience. The authors have responded effectively to reviewer concerns, particularly in demonstrating the necessity of the curvature metric and expanding their experimental comparisons to include additional baselines. They have also provided thorough explanations of the limitations and potential areas for future work, such as exploring class-separated representations further. While some concerns remain about the dependency on the quality of learned representations and the need for clearer guidance on parameter sensitivity, the overall contribution of the paper is significant. It introduces a new perspective on boundary detection in time series, backed by strong empirical evidence and theoretical insights. The method's robustness across different types of changes and datasets suggests that it has the potential to impact various applications significantly.